# Extinction recall of fear memories formed before stress is not affected despite higher theta activity in the amygdala

**Mohammed Mostafizur Rahman[1†], Ashutosh Shukla[1], Sumantra Chattarji[1,2,3]***

[1]National Centre for Biological Sciences, Bangalore, India; [2]Centre for Brain Development and Repair, Institute for Stem Cell Biology and Regenerative Medicine, Bangalore, India; [3]Centre for Integrative Physiology, Deanery of Biomedical Sciences, University of Edinburgh, Edinburgh, United Kingdom

**Abstract** Stress is known to exert its detrimental effects not only by enhancing fear, but also by impairing its extinction. However, in earlier studies stress exposure preceded both processes. Thus, compared to unstressed animals, stressed animals had to extinguish fear memories that were strengthened by prior exposure to stress. Here, we dissociate the two processes to examine if stress specifically impairs the acquisition and recall of fear extinction. Strikingly, when fear memories were formed before stress exposure, thereby allowing animals to initiate extinction from comparable levels of fear, recall of fear extinction was unaffected. Despite this, we observed a persistent increase in theta activity in the BLA. Theta activity in the mPFC, by contrast, was normal. Stress also disrupted mPFC-BLA theta-frequency synchrony and directional coupling. Thus, in the absence of the fear-enhancing effects of stress, the expression of fear during and after extinction reflects normal regulation of theta activity in the mPFC, not theta hyperactivity in the amygdala.
DOI: https://doi.org/10.7554/eLife.35450.001

*For correspondence:
shona@ncbs.res.in

Present address: †Department of Molecular and Cellular Biology, Center for Brain Science, Harvard University, Cambridge, United States

**Competing interests:** The authors declare that no competing interests exist.

## Introduction

Accumulating evidence from animal models shows that stress elicits divergent patterns of plasticity across brain regions (*Chattarji et al., 2015*). For instance, repeated stress causes loss of dendrites and spines in the medial prefrontal cortex (mPFC) (*Shansky and Morrison, 2009*). In the basolateral amygdala (BLA), by contrast, chronic stress strengthens the structural basis of synaptic connectivity through dendritic growth and spinogenesis (*Chattarji et al., 2015*). Physiological and molecular measures of synaptic plasticity also exhibit these contrasting features. As useful as these studies have been in examining the effects of stress across biological scales, much of this evidence was gathered from postmortem analyses (*Chattarji et al., 2015*). Less is known about how stress affects neural activity in the intact brain of behaving animals. Further, in many of these studies, stress-induced plasticity was viewed as stand-alone effects intrinsic to individual brain areas, despite extensive interconnections between them. Indeed, interactions between these brain areas together give rise to behaviors that are affected by stress (*Quirk and Mueller, 2008*).

One such behavior involves the expression of fear memories, various facets of which depend on both the BLA and mPFC (*Sierra-Mercado et al., 2011*). Repeated stress has been shown to enhance fear memories, as well as impair their extinction (*Miracle et al., 2006*; *Suvrathan et al., 2014*). These studies, however, first exposed animals to stress, and then subjected them to fear conditioning followed by extinction (*Izquierdo et al., 2006*; *Miracle et al., 2006*). In other words, these studies analyzed the impact of stress on both the formation *and* extinction of fear memories, not the latter alone. This experimental design has led to two broad classes of findings. In a majority of these studies, stress did not necessarily lead to higher freezing levels during fear recall. Even though

**eLife digest** Patients with stress-related psychiatric disorders experience debilitating emotional symptoms, including excessive fear that they are unable to control. Decades of research have shown that such disorders have opposite effects on two key structures in the brain. Normally, a region called the amygdala helps to form fear-related memories, while the prefrontal cortex helps control these memories and eliminate them through a process called extinction. But brain imaging on patients with stress-related psychiatric disorders reveals a smaller prefrontal cortex, and an overactive amygdala. Animal studies confirm that chronic stress shrinks brain cells in the prefrontal cortex, but grows them bigger in the amygdala.

These and other studies have led scientists to believe that stress causes people to both form stronger fear memories and then have difficulties getting rid of such memories. These include studies in which stressed animals were trained to fear a sound, and then followed while they learned to overcome that fear. One problem with many of these studies is that the animals were repeatedly stressed before the fear memory was formed. This made it hard to tease apart whether the strength of the fear memory itself or a problem extinguishing the fear memory were to blame for the animals' difficulties overcoming their fear.

Now, Rahman et al. address this problem by exploring how the timing of chronic stress affects how well an animal can overcome fear memories. In the experiments, some rats learned to fear a sound before exposure to chronic stress, while a second group was exposed to chronic stress first then learned to fear the sound. When the animals were stressed after they learned to fear the sound, they could still eliminate their fear. But the animals stressed before exposure to the fear-inducing sound struggled to extinguish the fear. Recordings of the brain activity in the rats that were exposed to stress after learning to fear the sound showed the amygdala remained overly active even after these animals overcame their fear. However, the stress did not seem to disrupt the normal activity of the prefrontal cortex in these rats. This shows that memories formed before stress reflect normal activity in the prefrontal cortex and not the abnormally high activity in the amygdala.

Exposure therapy helps people overcome stress-disorder related fears. For the therapy to work, the prefrontal cortex must be able to extinguish fear. Rahman et al. show that this is possible when the fear-enhancing effects of prior stress are not in play. More studies exploring why prior stress makes fears stronger and harder to overcome may help scientists develop ways to make therapies for stress disorders more effective.

DOI: https://doi.org/10.7554/eLife.35450.002

expression of fear was not elevated by prior exposure to stress, it was resistant to subsequent extinction in stressed animals (*Noble et al., 2017*; *Zhang and Rosenkranz, 2013*). In a few studies, however, the behavioral data suggest that prior exposure to stress may have also led to stronger fear memories manifested as higher levels of freezing in stressed animals at the beginning of extinction training (*Chauveau et al., 2012*; *Hoffman et al., 2014*; *Miracle et al., 2006*). This too could have contributed to the subsequent deficit in fear extinction recall. An alternative strategy to test if stress impairs the extinction of fear memories would be for animals to form fear memories *before* stress exposure, thereby dissociating the effects of stress on acquisition versus extinction of conditioned fear. This would offer an opportunity to examine how stress *specifically* affects expression of fear during and after extinction, without the confounds of stronger fear memories caused by prior exposure to stress. Hence, the present study combines simultaneous behavioral and in vivo electrophysiological analyses to address this question.

## Results

Rats were first subjected to auditory fear conditioning (Day 0, *Figure 1*) at the end of which they exhibited significantly higher freezing behavior in response to the tone conditioned stimulus (CS) (*Figure 1b*). These animals were then divided into two groups – one was subjected to 10 days of chronic immobilization stress (Days 1–10, *Figure 1a*) while the other served as unstressed control. 24 hr after the end of chronic stress there was no difference in CS-induced freezing behavior between the two groups (Day 11, Block1, *Figure 1b*). Thus, the recall of fear memory formed earlier was not

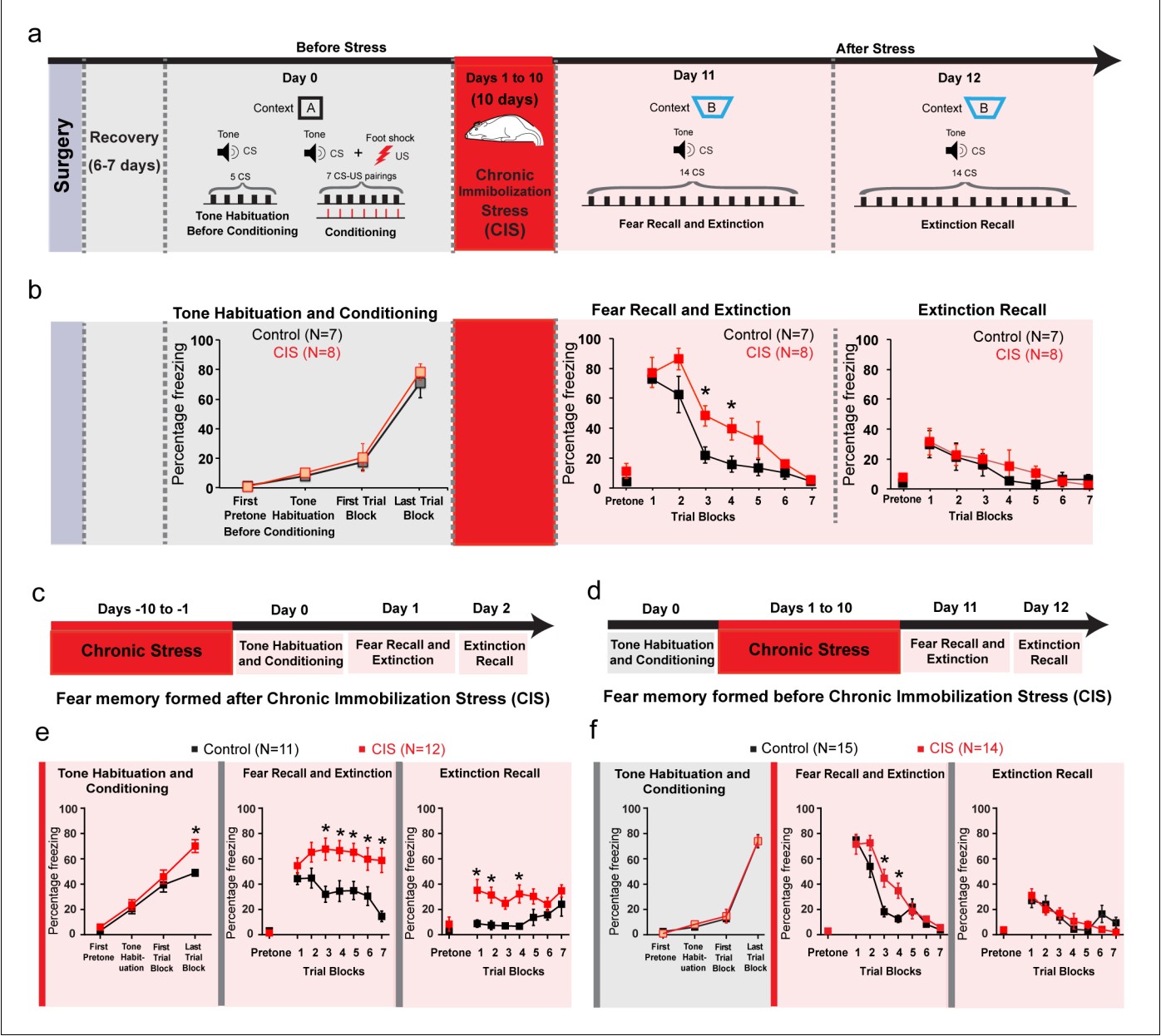

**Figure 1.** Normal expression of fear memories formed before stress. (**a**) Experimental design. Rats were subjected to surgery for implanting recording electrodes, and then allowed to recover for 6–7 days. Next the rats were subjected to tone habituation followed by fear conditioning on Day 0. 24 hr later some of these rats were subjected to chronic immobilization stress (CIS; 2 hr/d, 10d) while others were controls. Both groups underwent fear extinction 10 days later (Day 11), and extinction recall on Day 12. (**b**) Freezing at different time points. Significant increase in freezing relative to tone habituation after fear conditioning in both control and CIS groups together ($F_{(3,42)}$=91.71, p<0.01). No difference in fear recall at the start of the session between CIS (N = 8) and control (N = 7) rats (Day 11). CIS rats exhibited higher CS-induced freezing in the third and fourth trial blocks (p<0.05) indicating delay in acquisition of fear extinction. However, both groups eventually decreased freezing to the same level at the end of the 7 trial blocks (Factor: CIS $F_{(1,13)}$=4.85, p=0.05; Factor: Learning $F_{(6,78)}$=45.44, p<0.01; Factor: Interaction $F_{(6,78)}$=1.60, p=0.16). There was no difference in freezing between CIS and control animals during extinction recall (Factor: CIS $F_{(1,13)}$=0.14, p=0.72; Factor: Learning $F_{(6,78)}$=9.72, p<0.01; Factor: Interaction $F_{(6,78)}$=0.57, p=0.75). (**c–d**) Two different experimental designs to compare the effects of stress on fear recall and extinction by administering the same 10 day CIS either before (**c**) or after (**d**) the same fear conditioning protocol. (**c**) Animals were subjected to CIS from Day −10 to −1 followed by tone habituation and fear conditioning on Day 0. The animals were then subjected to fear recall and extinction on Day 1 followed by extinction recall on Day 2. (**d**) Animals were subjected to tone habituation and fear conditioning on Day 0 followed by CIS from Day 1 to 10. Next the animals were subjected to fear recall and extinction on Day 11 followed by extinction recall on Day 12. (**e**) When fear memory was formed after stress, the CIS animals (N = 12) show higher freezing compared to control animals (N = 11) only at the end of the conditioning session (Factor: CIS $F_{(1,21)}$=6.23, p=0.02; Factor: Learning $F_{(3,63)}$=75.15, p<0.01; Factor: Interaction $F_{(3,63)}$=2.48, p=0.07). Interestingly, both CIS and control groups show similar levels of freezing during fear recall

*Figure 1 continued on next page*

Figure 1 continued

at the start of the fear extinction session. However, the animals subjected to CIS fail to undergo extinction down to the same levels of freezing as seen in control animals (Factor: CIS $F_{(1,21)}$=15.23, p<0.01; Factor: Learning $F_{(6,126)}$=2.04, p=0.07; Factor: Interaction $F_{(6,126)}$=1.78, p=0.11). Notably, the CIS animals also exhibit significantly higher freezing during extinction recall relative to control rats (Factor: CIS $F_{(1,21)}$=15.64, p<0.01; Factor: Learning $F_{(6,126)}$=1.39, p=0.22; Factor: Interaction $F_{(6,126)}$=1.04, p=0.40). (**f**) Both control and CIS groups show learning induced enhancement in freezing ($F_{(3,84)}$=262.7, p<0.01). There was no difference in fear recall between CIS (N = 14) and control (N = 15) rats (Day 11). However, CIS rats exhibited higher CS-induced freezing in the third and fourth trial blocks (p<0.05) indicating delay in acquisition of fear extinction (Factor: CIS $F_{(1,27)}$=4.19, p=0.05; Factor: Learning $F_{(6,162)}$=71.05, p<0.01; Factor: Interaction $F_{(6,162)}$=4.13, p<0.01). But similar to the rats implanted with electrodes (**b**), both groups eventually reduced freezing to the same level at the end of the 7 trial blocks. There was no difference in freezing behavior during extinction recall (Factor: CIS $F_{(1,27)}$=0.03, p=0.86; Factor: Learning $F_{(6,162)}$=14.67, p<0.01; Factor: Interaction $F_{(6,162)}$=2.28, p=0.04). Data are mean ±s.e.m. in blocks of two trials except pretone. *p<0.05.

DOI: https://doi.org/10.7554/eLife.35450.003

The following source data is available for figure 1:

**Source data 1.** Data for animals across groups representing freezing response to CS and pretone during the different phases of behaviour (**Figure 1b, e and f**).

DOI: https://doi.org/10.7554/eLife.35450.004

affected by subsequent stress. This ensured that both stressed and unstressed animals were at the same levels of freezing when the extinction protocol was initiated after stress. Next, repeated tone presentations reduced freezing levels significantly such that both groups eventually underwent comparable extinction of fear, though the stressed rats were slower in achieving the same reduction in freezing (Day 11, **Figure 1b**). Notably, a day later stressed animals showed no difference in recall of extinction memory compared to unstressed animals (Day 12, **Figure 1b**). Thus, freezing levels during both fear and extinction recall were unaffected in stressed animals. This result differs from past reports of stress-induced deficits in fear extinction. However, as mentioned earlier, those earlier studies subjected animals to fear conditioning and extinction *after* exposure to stress (**Izquierdo et al., 2006**; **Maren and Holmes, 2016**; **Miracle et al., 2006**). Taken together, this suggests that the *timing* of stress may be a critical determinant of whether extinction recall is impaired. Thus, to examine this possibility we repeated the experimental design adopted in earlier studies by administering the same 10 day chronic immobilization stress protocol before (**Figure 1c,e**) the same fear conditioning paradigm depicted in **Figure 1a**. Consistent with earlier findings, prior exposure to chronic stress caused a significant impairment in the recall of fear extinction (**Figure 1e**). However, these animals, unlike those used in **Figure 1b** (i.e. conditioning before stress) were not implanted with electrodes for simultaneous in vivo recordings. Hence, we repeated the behavioral experiments described in **Figure 1b** without surgical interventions related to in vivo recordings. This too yielded the same results as seen in the implanted animals – extinction recall was intact in stressed animals (**Figure 1d,f**) when they were subjected to conditioning before chronic stress.

Next, we examined the neural basis of this result by recording local field potentials (LFPs) in these freely behaving rats (**Karalis et al., 2016**; **Likhtik et al., 2014**). While a role for potentiation of amygdalar neuronal responses to the tone CS in conditioned fear behavior is well established, in vivo recordings have also shown correlations of tone responses in the dorsal mPFC (dmPFC) with freezing behavior in fear conditioning and extinction. Taken together with earlier pharmacological inactivation studies, these findings identified an important role for the dmPFC in underlying conditioned fear responses and the expression of fear extinction (**Burgos-Robles et al., 2009**; **Sierra-Mercado et al., 2011**). Therefore, in addition to the BLA, we also monitored responses in the dmPFC (**Figure 2a**, **Figure 2—figure supplement 1**). We first analyzed CS-evoked LFPs in the BLA at three key behavioral time points described in **Figure 1** – tone habituation before conditioning, fear recall and extinction recall (**Figure 2b–d**). We measured auditory evoked potential (AEP) amplitudes as the difference between first peak and first trough in the AEP (**Figure 2—figure supplement 1**). During fear recall, AEP amplitudes (**Rogan et al., 1997**) were enhanced in both stressed and unstressed animals (**Figure 2c,e**). However, while this increase was reversed in unstressed animals, it persisted in stressed rats even during extinction recall (**Figure 2c,e**). Previous work also identified increase in CS-evoked theta power as a neural correlate of conditioned fear (**Likhtik et al., 2014**). We found BLA theta power (measured as the power of auditory evoked responses in the 2–12 Hz frequency band) in unstressed animals also paralleled the increase, followed by decrease, in freezing during fear and extinction recall

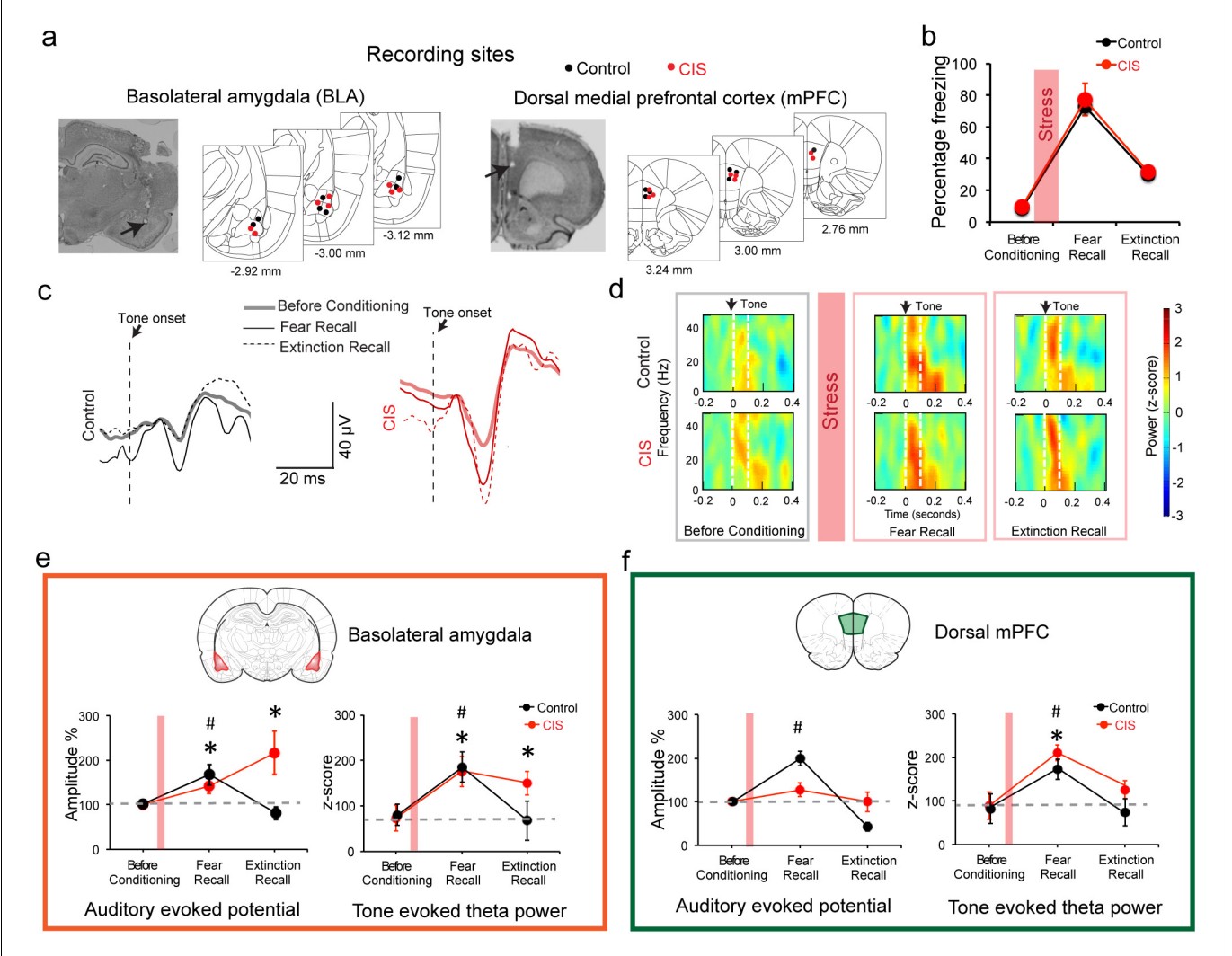

**Figure 2.** Stress-triggered persistent hyperactivity in the BLA but not the dmPFC. (**a**) Representative micrographs and diagrams of recording electrode placements (red: CIS, black: Control) in the BLA (*left*) and dmPFC (*right*). (**b**) Summary of changes in tone-induced freezing behavior before conditioning, and during fear and extinction recall. Stress (red bar) did not affect fear retrieval or extinction (Factor: CIS $F_{(1,13)}$=0.26, p=0.62; Factor: Learning $F_{(2,26)}$=51.95, p<0.01; Factor: Interaction $F_{(2,26)}$=0.02, p=0.98). (**c**) Representative raw LFP traces depicting changes in AEPs recorded in the BLA in response to CS presented during habituation before conditioning, fear and extinction recall in animals subject to CIS (red) and unstressed control (black) animals. (**d**) Representative spectrograms of BLA LFPs before conditioning (*left*), fear recall (*center*) and extinction recall (*right*) recorded from control (top) and CIS (bottom) animals. Dotted white lines on the spectrogram indicate onset (arrow) and end of CS. (**e**) Percentage changes (normalized to tone habituation before conditioning) in auditory evoked potential amplitude (AEP, *left*) and auditory evoked theta power (*right*) in the BLA. All statistical comparisons are done within groups across the three time points, not between CIS (N = 8) and control (N = 7) rats. In unstressed control animals, AEP amplitudes increased during fear recall compared to tone habituation ($F_{(2,12)}$=7.39, p<0.01) and was subsequently reversed during extinction recall. In CIS animals, by contrast, BLA AEP amplitudes were enhanced during both fear recall and extinction recall ($F_{(2,14)}$=5.78, p=0.01). The same pattern of changes were observed in BLA theta power during fear recall in control animals ($F_{(2,12)}$=5.66, p=0.02), while CIS animals exhibited theta power enhancement during fear recall that persisted even during extinction recall ($F_{(2,14)}$=13.70, p<0.01). (**f**) Percentage changes (normalized to tone habituation before conditioning) in AEPs (*left*) and auditory evoked theta power (*right*) in the dmPFC. All statistical comparisons are done within groups across the three time points, not between CIS (N = 8) and control (N = 7) rats. AEP amplitudes in control animals increased only during fear recall ($F_{(2,12)}$=62.01, p<0.01) whereas the animals subjected to CIS exhibited no changes ($F_{(2,14)}$=1.59, p=0.24). But dmPFC theta power increased in both control and CIS animals during fear recall relative to tone habituation (Control $F_{(2,12)}$=5.89, p=0.02; CIS $F_{(2,14)}$=11.15, p<0.01). And these increases were reversed during extinction recall in both groups. Data are mean ±s.e.m. in each block; #p<0.05, Control animals; *p<0.05, CIS animals.
DOI: https://doi.org/10.7554/eLife.35450.005

The following source data and figure supplements are available for figure 2:

**Source data 1.** Data for animals representing freezing response to CS (Figure 2b) and AEP amplitude and theta power in dmPFC and BLA (*Figure 2e,f*)

*Figure 2 continued on next page*

*Figure 2 continued*

DOI: https://doi.org/10.7554/eLife.35450.008

**Source data 2.** Data for animals representing freezing response to CS (*Figure 2—figure supplement 2b*) and AEP amplitude and theta power in dmPFC and BLA (*Figure 2—figure supplement 2c,d*)

DOI: https://doi.org/10.7554/eLife.35450.009

**Figure supplement 1.** Representative traces and power spectra of tone evoked local field potentials (LFPs).

DOI: https://doi.org/10.7554/eLife.35450.006

**Figure supplement 2.** Theta activity in the dmPFC, but not the BLA, mirrors the gradual decrease in freezing levels during the acquisition of fear extinction.

DOI: https://doi.org/10.7554/eLife.35450.007

respectively (*Figure 2d–e*). By contrast, BLA theta power remained high in stressed animals (*Figure 2d–e*). Thus, despite stress-induced theta hyperactivity in the BLA, fear expression was not enhanced during extinction recall. To probe this further, we also analyzed the same LFP parameters in the dmPFC, which according to recent studies plays an important role in fear expression (*Karalis et al., 2016*; *Likhtik et al., 2014*). In the dmPFC of unstressed rats, AEP amplitude increased during fear recall, and this was reversed during extinction recall (*Figure 2f*). Stressed animals, however, did not exhibit any change in dmPFC AEP amplitudes during either fear or extinction recall. Further, fear conditioning enhanced dmPFC theta power in both stress and unstressed animals (*Figure 2f*). Interestingly, this was reversed in both groups during extinction recall. In other words, unlike the BLA, changes in theta power in the dmPFC, during fear and extinction recall, were not affected by stress. Moreover, in stressed animals, bidirectional modulation of theta power in the dmPFC, but not the BLA, accurately mirrored the changes in freezing, a behavioral expression of fear. Interestingly, the gradual decrease in dmPFC theta power paralleled the significant within-session reduction in freezing during the acquisition of extinction (Day 11, *Figure 1b*) in both control and stressed animals (*Figure 2—figure supplement 2b,d*). However, unlike the slower time course of extinction in the stressed rats, there was no difference in the time course of reduction in dmPFC theta power between stressed versus control animals. Similar analysis of BLA theta power during extinction learning (Day 12, *Figure 1b*) did not reveal any significant differences that paralleled the gradual within-session decrease in freezing levels (*Figure 2—figure supplement 2b,c*).

Finally, there is growing appreciation of the importance of interactions between the mPFC and BLA, not just activity within these areas, in regulating fear behavior (*Karalis et al., 2016*; *Lesting et al., 2011*; *Likhtik et al., 2014*; *Popa et al., 2010*). This issue comes into sharp focus here because of the distinct effects of stress on the mPFC versus BLA. Thus, in light of recent reports that theta frequency oscillations synchronize dmPFC–BLA circuits during expression of fear behavior (*Karalis et al., 2016*; *Likhtik et al., 2014*), we investigated whether the tone-evoked increases in theta power (*Figure 2*) were accompanied by enhanced theta-frequency synchrony between the two areas, and if this was in anyway affected by stress. Hence, we quantified CS-evoked coherence to assess moment-by-moment synchrony across LFPs recorded from the dmPFC and BLA for all three time points (*Figure 3a*) (*Likhtik et al., 2014*). In unstressed rats, consistent with earlier reports, the CS elicited significantly higher theta coherence during fear recall (*Likhtik et al., 2014*), and this increase persisted during extinction recall as well (*Figure 3b*). Notably, in stressed animals, there was no change in BLA-dmPFC theta-frequency coherence (*Figure 3b*). Thus, stress appears to completely suppress the dynamic, behaviorally relevant enhancement in BLA-dmPFC coherence that is seen during both fear and extinction recall in unstressed animals. In light of strong reciprocal connections between the mPFC and BLA, increases in theta synchrony have led earlier studies to analyze the direction of information flow between the two areas (*Karalis et al., 2016*; *Likhtik et al., 2014*). Hence, we estimated the directionality of functional connectivity and leads between the dmPFC and BLA using a previously validated method of calculating cross-correlations of instantaneous amplitude of filtered LFPs (*Adhikari et al., 2010*). This reveals that theta activity in the dmPFC leads that in the BLA during recall of both fear and extinction memories in unstressed rats (*Figure 3c*). However, this dmPFC-to-BLA directional influence is absent in stressed animals. Together, these data suggest that chronic stress causes a decoupling of activity between the two brain areas, as evidenced by a complete disruption of the increase in dmPFC-BLA theta synchrony and dmPFC-to-BLA directional influence normally seen during the recall of fear and extinction memories. This decoupling is also evident across all trial blocks

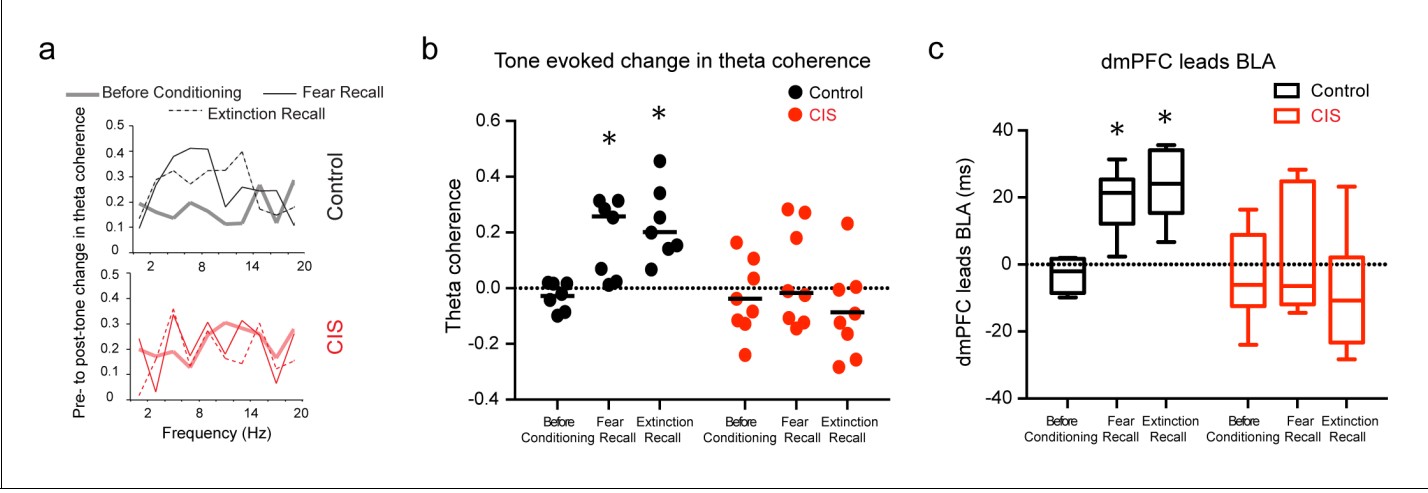

**Figure 3.** Stress disrupts enhanced dmPFC-BLA theta synchrony and directional coupling during fear expression. (a) Tone-evoked changes in theta-frequency coherence between the BLA and dmPFC in exemplars of control (*top*) and CIS (*bottom*) animals. Control animal exhibited higher synchrony during fear recall and extinction recall than tone habituation. These changes were absent in the rat subjected to CIS. (b) Means and distribution of tone-evoked changes in theta-frequency coherence for both groups (Control N = 7, CIS N = 8). The control animals show enhanced theta coherence during fear recall and extinction recall ($F_{(2,12)}$=10.81, p<0.01). But animals subjected to CIS did not show any change in theta coherence across sessions ($F_{(2,14)}$=1.01, p=0.39). Data are presented as scatter plots with medians. (c) Estimation of leads between the dmPFC and BLA using the amplitude cross-correlation. The dmPFC leads more than chance over the BLA during fear and extinction recall in control (N = 7), but not CIS (N = 8) rats. Data are presented as medians ± maxima/minima. *p<0.05 significantly different from chance for each time point in each group.

DOI: https://doi.org/10.7554/eLife.35450.010

The following source data and figure supplement are available for figure 3:

**Source data 1.** Data for animals representing theta coherence between dmPFC and BLA (*Figure 3b*) and dmPFC-BLA phase difference (*Figure 3c*)
DOI: https://doi.org/10.7554/eLife.35450.012
**Source data 2.** Data for animals representing theta coherence between dmPFC and BLA (*Figure 3—figure supplement 1b*) and dmPFC-BLA phase difference (*Figure 3—figure supplement 1c*)
DOI: https://doi.org/10.7554/eLife.35450.013
**Figure supplement 1.** Stress also disrupts dmPFC-BLA theta synchrony and directional coupling during the acquisition of fear extinction.
DOI: https://doi.org/10.7554/eLife.35450.011

during the acquisition of extinction the previous day (*Figure 3—figure supplement 1*). This breakdown in mPFC-BLA theta-frequency synchrony and directional coupling could be one reason why enhanced BLA theta activity was not manifested as higher freezing in stressed animals.

The present study, specifically designed to administer chronic stress *after* the formation of fear memory, reveals that when stressed animals started extinguishing fear memories from the same level of freezing as their unstressed counterparts, their ability to recall extinction memory remained intact. This is in contrast to past findings wherein stressed animals exhibited a deficit in extinction recall when faced with the challenge of extinguishing fear memories strengthened by prior exposure to stress. This is consistent with earlier findings that repeated immobilization stress (4 h/day for 14 days) administered after auditory fear conditioning did not affect the expression of previously acquired fear memories (*Meyer et al., 2014*). However, the same immobilization stress, when administered *before* fear conditioning, elicited a robust increase in fear recall. Together, these results suggest that the timing of chronic stress exposure, with respect to fear conditioning, is a key determinant of how subsequent extinction of fear is affected by stress. This dissociation of the effects of stress on acquisition versus extinction of fear conditioning suggests that stress acts primarily on acquisition, and the earlier findings from pre-conditioning stress experiments were not the result of a deficit in fear extinction per se. Age may also influence how stress affects fear extinction in rats. For instance, deficient fear extinction was only observed in previously stressed adolescent, but not adult, rats (*Zhang and Rosenkranz, 2013*).

Interestingly, despite no visible behavioral effect of stress on fear expression, our in vivo recordings reveal a robust impact of stress on amygdalar activity, as evidenced by enhanced theta activity

in the BLA that failed to reverse even after the animals exhibited normal extinction recall (*Figure 2e*). This is consistent with earlier findings on physiological and structural strengthening of excitatory synaptic connectivity, as well as reduced inhibitory tone, in the BLA after stress (*Suvrathan et al., 2014*). Together, these changes point to possible synaptic mechanisms in the BLA for the stress-induced strengthening of subsequent encoding of fear memories.

The same chronic stress, on the other hand, did not disrupt normal bidirectional modulation of mPFC theta activity, which in turn was reflected in normal freezing behavior during recall of fear and extinction (*Figure 2f*). This is consistent with growing evidence for a pivotal role played by the mPFC in fear expression (*Dejean et al., 2016*; *Sierra-Mercado et al., 2011*). For instance, recent work has demonstrated strong correlations between mPFC theta-frequency oscillations and conditioning-induced freezing behavior (*Likhtik et al., 2014*). Furthermore, we find normal mPFC theta activity to be decoupled from theta hyperactivity in the BLA, possibly reducing the latter's influence on fear expression. This is similar to a report that even a single episode of stress can weaken functional connectivity between the two areas measured by resting state fMRI (*Liang et al., 2014*). Indeed, such stress-induced disruptions in prefrontal-to-amygdala connectivity is also known to affect social interaction and anxiety-related behaviors in rodents (*Adhikari et al., 2015*; *Hultman et al., 2016*).

Our findings, taken together with earlier behavioral studies (*Izquierdo et al., 2006*; *Meyer et al., 2014*; *Miracle et al., 2006*; *Rau and Fanselow, 2009*; *Sierra-Mercado et al., 2011*; *Suvrathan et al., 2014*) suggest a model for why the effects of stress on the recall and extinction of fear memories depend on the timing of stress exposure with respect to when the fear memory is formed and extinguished (*Figure 4*). Specifically, these findings suggest that the regulation of remote fear memories by stress (i.e. pre-stress conditioning, *Figure 4b*) is different from that of more recent fear memories (i.e. post-stress conditioning, *Figure 4c*). As reported here, when fear memories were formed *before* exposure to the 10 day chronic stress, recall of fear extinction was not affected. Although stress caused a persistent increase in theta activity in the BLA, it did not affect bidirectional regulation of dmPFC theta activity (*Figure 4b*). While stress-induced disruption in mPFC-BLA directional coupling and theta-frequency synchrony (*Figure 3*) may explain why enhanced BLA theta activity did not result in stress-induced enhancement in freezing, another intriguing possibility arises from a recent report that neural circuits mediating the recall of fear memories shifts over time (*Do-Monte et al., 2015*). Specifically, this study showed that optogenetic silencing of BLA reduced freezing when it was performed 6 hr, but not 7 days, after auditory fear conditioning. Since our post-conditioning stress protocol was carried out for 10 days, it may have outlasted BLA involvement in expression of fear and could also explain why enhanced BLA excitability was no longer reflected in freezing at that remote time point (Day 11; *Figure 4b*). As a first step towards testing this possibility, we carried out in vivo infusions of muscimol directly into the BLA just before fear recall and extinction learning on Day 11 (*Figure 4—figure supplement 1*). Strikingly, this targeted inactivation of the BLA had no impact on fear expression at this time point in either stressed or control rats (*Figure 4—figure supplement 1b*). Further, freezing levels in the muscimol-infused control and stressed rats did not differ from the freezing levels seen in stressed/control animals (*Figure 4—figure supplement 1c*) that did not receive any infusions in earlier experiments (*Figure 1f*). Thus, we found that in vivo inactivation of the BLA no longer affects the recall of fear memories that were formed before the 10 day stress exposure. While more detailed analyses will be required to test all aspects of this model, these preliminary results suggest that fear memories formed 10 days ago no longer depend on BLA activity in either control or stressed animals. Consequently, the expression of fear after extinction reflects normal regulation of theta activity in the mPFC, not theta hyperactivity in the BLA (*Figure 4b*).

On the other hand, when fear memories were formed *after* exposure to the same chronic stress, the behavioural effects on freezing are very different, as has also been reported in earlier studies that employed a post-stress conditioning strategy (*Maren and Holmes, 2016*; *Miracle et al., 2006*; *Rau and Fanselow, 2009*). Animals that were fear conditioned after stress show enhanced freezing during both acquisition and recall of extinction memory. What could be the neural basis for this difference? Electrophysiological data from our post-conditioning stress experiments point to possible mechanisms that can be examined in future studies (*Figure 4c*). For instance, in contrast to the pre-stress conditioning situation, stressed animals will recall and extinguish fear memories that were formed only 1 day ago. As a result, stress-induced BLA hyperactivity will be manifested as stronger

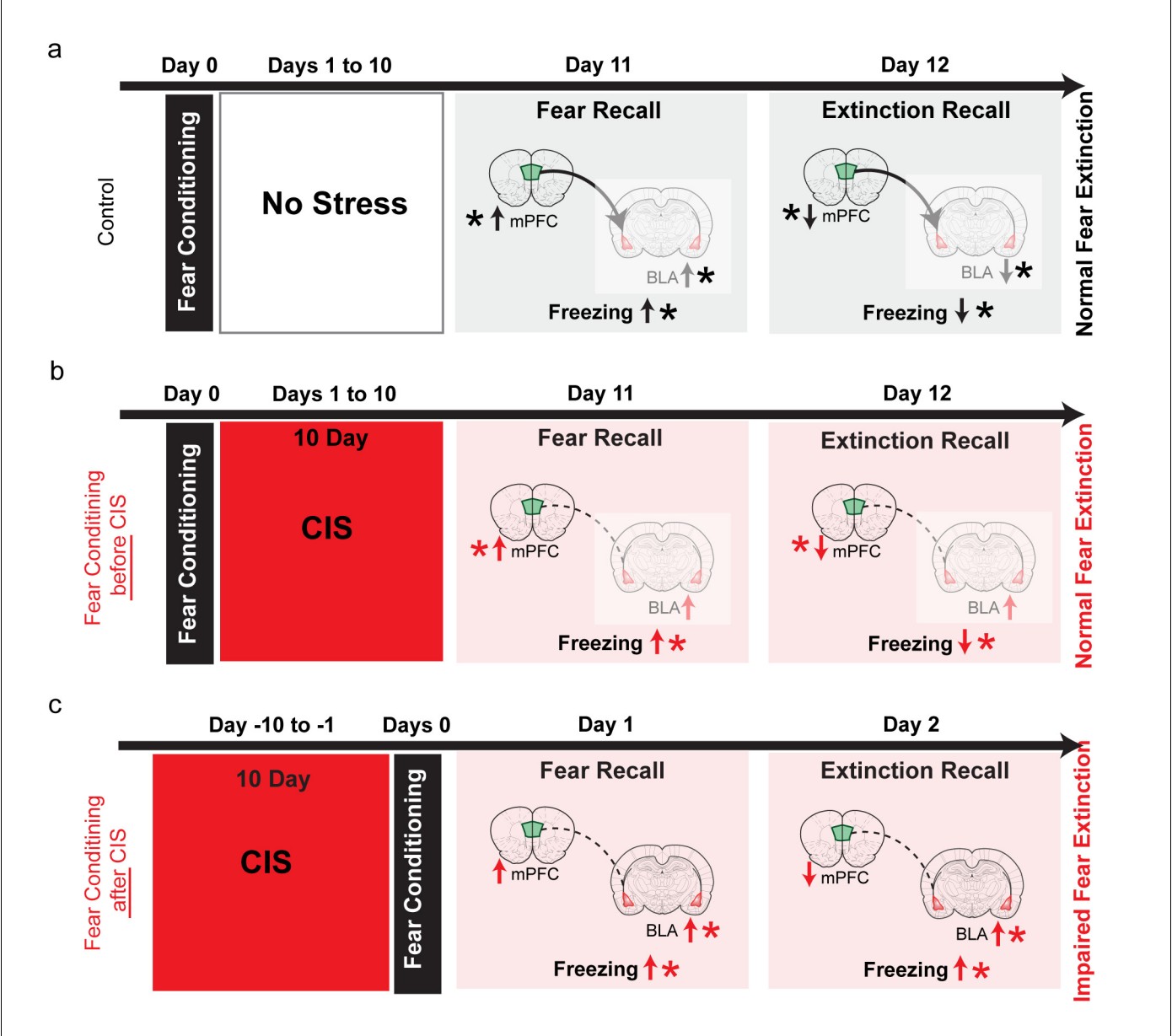

**Figure 4.** Summary of results and proposed model for how the effects of chronic stress on extinction recall depend on whether the fear memory was formed before or after stress. (a) In control animals, fear conditioning enhances freezing, which is subsequently reduced by extinction (black vertical arrows). This is accompanied by, first an increase and then a reduction, in theta activity in both the BLA and dmPFC (black vertical arrows). Also, stronger dmPFC-BLA theta synchrony and dmPFC-to-BLA directional influence (thicker curved line with arrow from dmPFC to BLA) is seen during fear and extinction recall. Thus, the direction of changes in freezing are mirrored by in vivo electrophysiological changes in both the BLA and mPFC (indicated by *), although 10 days after conditioning fear expression no longer depends (faded colors) on BLA activity (*Do-Monte et al., 2015*), *Figure 4—figure supplement 1*). (b) When fear memories were formed *before* exposure to chronic immobilization stress (CIS; 2 hr/d for 10d), recall of fear extinction was not affected (red vertical arrows). Although theta activity in the BLA exhibited a persistent increase, its bidirectional regulation was normal in the mPFC (red vertical arrows). CIS also disrupted mPFC-BLA theta-frequency synchrony and directional coupling (dotted lines). However, fear memories formed 10 days ago no longer depend on BLA activity (faded colors), (*Do-Monte et al., 2015*), *Figure 4—figure supplement 1*). Consequently, the expression of fear after extinction reflects normal regulation of theta activity in the mPFC (indicated by *), not theta hyperactivity in the BLA. (c) Consistent with earlier behavioral studies, when fear memories were formed *after* exposure to the same CIS, the behavioural effects on freezing are strikingly different. Stressed animals show enhanced freezing during both acquisition and recall of extinction memory (red vertical arrows). Based on these behavioral findings, and electrophysiological data from our post-conditioning stress experiments (shown in panel b), we hypothesize that CIS will cause a sustained increase in BLA theta activity, but not affect normal bidirectional regulation of dmPFC theta activity (red vertical arrows). Unlike the pre-stress conditioning situation, the CIS animals are now recalling and extinguishing fear memories that were formed only 1 day ago. As a result, CIS-induced BLA hyperactivity will be manifested as stronger fear memories that are resistant to subsequent extinction (indicated by higher

*Figure 4 continued on next page*

*Figure 4 continued*

freezing during extinction recall in CIS rats). Thus, now it is the sustained increase in BLA theta activity, not normal regulation of dmPFC activity, that better reflects higher freezing in CIS rats during extinction learning as well as extinction recall (indicated by *). Moreover, CIS-induced disruption of mPFC-BLA theta-frequency synchrony and directional coupling (dotted lines) is likely to impair the regulation of BLA activity by the mPFC.

DOI: https://doi.org/10.7554/eLife.35450.014

The following source data and figure supplement are available for figure 4:

**Source data 1.** Data for animals across groups representing freezing response to CS during the different phases of behaviour (*Figure 4—figure supplement 1c,d*)

DOI: https://doi.org/10.7554/eLife.35450.016

**Figure supplement 1.** Preliminary results on how pharmacological inactivation of the BLA has no impact on recall of fear memories formed before stress.

DOI: https://doi.org/10.7554/eLife.35450.015

fear memories that are resistant to subsequent extinction (as evidenced by higher freezing during extinction recall in CIS rats, *Figure 1e*). Thus, now it is the sustained enhancement in BLA theta activity, not normal regulation of dmPFC activity, that is likely to drive higher freezing in stressed rats during extinction learning and extinction recall (*Figure 4c*). Further, the detrimental effects of stress will be compounded by the disruption of mPFC-BLA theta-frequency coherence and directional coupling, which is likely to impair the mPFC's ability to regulate BLA hyperactivity. Interestingly, pre-conditioning stress has been shown to cause subsequent enhancement in fear learning that can last up to 90 days later (*Rau and Fanselow, 2009*). This naturally raises questions about the involvement of the BLA in the enhanced expression of conditioned fear at remote time points. Our preliminary findings (*Figure 4—figure supplement 1*), taken together with the study by *Do-Monte et al. (2015)*, suggest that despite the effects of pre-conditioning stress on the *encoding* of fear memories, subsequent *expression* at later time points may no longer depend on BLA activity. This, in turn, raises the possibility that other brain areas that interact with the BLA in the regulation of fear, such as the dmPFC, may play a role in maintaining this expression at remote time points. Thus, a better understanding of the neural basis of earlier behavioral findings (*Izquierdo et al., 2006*; *Meyer et al., 2014*; *Miracle et al., 2006*; *Noble et al., 2017*; *Rau and Fanselow, 2009*) will require detailed in vivo electrophysiological analyses, coupled with targeted inactivation, to examine the role of other brain areas that are engaged in stress-induced changes of fear expression.

Finally, it is interesting to note that the chronic immobilization stress paradigm used in the present study was previously shown to strengthen functional connectivity from the amygdala to the hippocampus (*Ghosh et al., 2013*), which undergoes stress-induced deficits similar to the mPFC (*Arnsten, 2015*; *Chattarji et al., 2015*; *McEwen and Morrison, 2013*). In other words, although both the hippocampus and mPFC undergo similar forms of stress-induced deficits, the impact of stress on their individual interactions with the amygdala are strikingly different. A better understanding of these divergent features of aberrant interactions distributed across the amygdala-mPFC-hippocampal network, not just those confined within each area, may offer new insights into therapeutic interventions against the cognitive and emotional symptoms of stress-related psychiatric disorder.

## Materials and methods

### Experimental animals

Naïve 8–9 weeks old male Sprague-Dawley rats (RRID: RGD_734476) weighing 300–350 grams at the start of the experiment (National Centre for Biological Sciences, Bangalore, India) and housed in groups of two were used in the study. They were maintained on a 14 hr/10 hr light/dark cycle and had access to water and a standard diet *ad libitum*. All experiments were conducted in accordance with the guidelines of the CPCSEA, Government of India and approved by the Institutional Animal Ethics Committee of National Centre for Biological Sciences.

### Experimental design

The experimental design comprised of experimental procedures conducted over a 4 week period. The animals were handled for 2–3 days to familiarize with the experimenter. This was followed by a surgery

to implant bundle of electrodes in the basolateral amygdala and the dorsal medial prefrontal cortex of the animals. The animals were allowed to recover for 6–7 days after surgery. Next, the animals were subjected to a 15 day behavioral paradigm with simultaneous recording of local field potentials (LFPs) during behavior (*Figure 1a*). The animals were initially habituated to the conditioning context on Day −1 and Day 0. Next on Day 1 the animals were subjected to the tone habituation and fear conditioning protocol. Then on Day 2 the animals were randomly allotted to the chronic immobilization stress (CIS) or control groups. The animals in the CIS group were subjected to a 10 day chronic immobilization stress (CIS; 2 h/day for 10 consecutive days) from Day 2 to day 11, whereas the animals in the control group were just handled once a day during the same period. Subsequently, on Day 12, that is 24 hr after the end of CIS, the animals were subjected to fear recall and extinction training. On Day 13, the animals were subjected to fear extinction recall session. After the end of the behavioral paradigm the animals were sacrificed and brains were collected for histological examination.

For the behavior experiments without LFP recordings, the animals were handled for 2–3 days and then subjected to the same behavior protocol described above. For assessing fear memory formed prior to stress, the animals were subjected to tone habituation and conditioning on day 1 followed by the 10 day CIS paradigm. Subsequently, the animals were subjected to fear recall and extinction on Day 12 and extinction recall on Day 13. For assessing fear memory formed after stress, the animals were first subjected to the CIS paradigm followed by tone habituation and conditioning on Day 11. Subsequently, the animals were subjected to fear recall and extinction training on Day 12 and extinction recall on Day 13. The animals were randomly allocated to either CIS or control groups. The animals were pair-housed and stressed and unstressed animals were separately housed.

For the behavior experiments that did not involve BLA inactivation prior to fear recall (see details below), the animals were handled for 2–3 days followed by surgery to implant stainless steel cannulae (24 gauge, Plastic One, Roanoke, Virginia, USA) targeted at the basolateral amygdala (BLA). Following the surgery, animals were single housed and allowed to recover for 6–7 days following the surgery. Next, the animals were subjected to tone habituation followed by fear conditioning on Day 1. The animals were then randomly split into CIS and control groups. CIS rats were subjected to the same 10 day chronic immobilization stress paradigm. Subsequently, on Day 12 all the animals received intracranial infusion of muscimol into the BLA. 30 min after the muscimol infusion the animals were subjected to fear recall and extinction session. Finally, the animals underwent extinction recall on Day 13.

## Surgical procedure

For recording LFPs from the dmPFC and BLA, rats were surgically implanted with formavar insulated nichrome wire (25 microns diameter; AM Systems, Carlsborg, WA, USA) bundles unilaterally in the BLA and dmPFC. And for the BLA inactivation experiments the animals were implanted with stainless steel cannulae (24 gauge, Plastic One, Roanoke, Virginia, USA) targeting the BLA bilaterally.

Rats were induced into anesthesia with 5% isoflurane (Forane, Asecia Queensborough, UK) and then maintained in anesthesia with 1.5–2% isoflurane. The level of anesthesia was regularly monitored throughout the procedure using the pedal withdrawal reflex to toe pinch. The animal was placed and head fixed on a stereotaxic frame. Body temperature of rats was maintained with a heating pad. For implanting electrode bundles, burr holes were drilled at the stereotactic coordinates of the BLA (stereotaxic coordinates were: 3.0 mm posterior to bregma and ±5.3 mm lateral to midline (*Paxinos and Watson, 2009*) and the dorsomedial prefrontal cortex (dmPFC; stereotaxic coordinates were: 3.0 mm anterior to bregma and ±0.5 mm lateral to midline, *Paxinos and Watson, 2009*). A bundle of 8 formavar coated nichrome electrodes were then implanted using the stereotactic frame (8.3 mm and 3.4 mm ventral from the brain surface for BLA and dmPFC respectively). Head screws were implanted to anchor the implant. One head screw placed just behind the lambda on the skull was used as the ground electrode. For implanting cannulae, burr holes were drilled at the stereotactic coordinated of the BLA bilaterally (stereotaxic coordinates were: 3.0 mm posterior to bregma and ±5.2 mm lateral to midline (*Paxinos and Watson, 2009*). Stainless steel guide cannulae (24 gauge, Plastic One, Roanoke, Virginia, USA) were then implanted using the stereotactic frame (7 mm ventral from the brain surface). Dummy cannulae (28 gauge, Plastic One, Roanoke, Virginia, USA) with 0.5 mm projection were inserted into the cannulae to prevent clogging. The implant was secured using anchor screws and dental acrylic cement. Rats were allowed to recover for 6–7 days following surgery. In the post-surgery period the animals were singly housed in separate cages.

A total of 18 animals were implanted with electrodes. Three animals were excluded from the study because the positioning of the electrode bundles was incorrect. The location of the cannulae placement for the 15 animals used in the LFP experiments is shown in *Figure 2a*. A total of 16 animals were implanted with cannulae and one was excluded from the study due to an error with muscimol infusion.

## Stress protocol

Rats in the CIS group were subjected to a chronic immobilization stress (CIS) paradigm (*Ghosh et al., 2013*), consisting of complete immobilization for 2 hr per day (before noon) in rodent immobilization bags without access to either food or water, for 10 consecutive days.

## Fear conditioning and extinction training protocol

Fear conditioning and extinction took place in different contexts placed inside sound-isolation boxes (Coulbourn Instruments, Whitehall, Pennsylvania, USA). Conditioning was performed in a box with metal grids on the floor (context A: 12 inches wide ×10 inches deep ×12 inches high, no odour). Fear extinction training and extinction recall was performed in another context, a modified home-cage (context B: 14 inches wide ×8 inches deep ×16 inches high, mint odour). Lighting conditions and walls were different between the two contexts. All chambers were cleaned with 70% alcohol before and after each experiment.

The behavior of the animals was recorded using a video camera mounted on the wall of the sound isolation box and a frame grabber (sampled at 30 Hz). The videos were analyzed offline for further quantification of freezing behavior. Infrared LED cues were placed on the walls of the experimental chambers. These cues were activated in coincidence with auditory stimuli to monitor the tone-evoked freezing response offline. A programmable tone generator and shocker (Habitest system, Coulbourn Instruments, Whitehall, Pennsylvania, USA) were used to deliver tones and foot-shocks during the experiment. Foot-shocks were delivered through the metal grids on the floor of the conditioning chamber. The tone was played using a speaker (4 Ω, Coulbourn Instruments, Whitehall, Pennsylvania, USA) placed inside the experimental chamber.

During context habituation, the animals were allowed to explore context A for 25 min in each session. Next, in the tone habituation session (*Figure 1a*) the animals received five presentations of an auditory tone (total duration of 30 s, 5 kHz auditory tone consisting of 30 pips of 100 millisecond duration at a frequency of 1 Hz; 5 millisecond rise and fall, 70 ± 5 dB sound pressure level) in context A. This was immediately followed by fear conditioning protocol, where the tone (CS) was paired ( pairings, average inter-trial interval <ITI > =120 s, with a range of 80–160 s) with a co-terminating 0.5 s scrambled foot shock (US; 0.7 mA). In the fear recall and extinction session the animals were presented with the same tone (CS) for 15 times (average inter-trial interval <ITI > =120 s, with a range of 80–160 s) in the context B. Again, in the extinction recall session, the animals were subjected to the same CS 15 times again.

## Behavioral analysis

Behavioral response was scored offline using video recordings of all the behavior sessions. Response to the auditory stimuli was evaluated in the form of freezing response. Freezing was defined as the absence of movement except due to respiration(*Blanchard and Blanchard, 1988*). The time spent freezing during the presentation of the tone was converted into a percentage score (*Figure 2c*). The percentage freezing level was measured in every context/session for 30 s immediately before the presentation of the first tone trial to assess freezing in absence of an auditory stimulus. This was defined as the freezing in the pretone period (*Figure 2d*). The pretone block represents freezing during the first pretone only. The tone habituation block represents freezing during the last two trials of tone habituation. The first and last trial blocks during conditioning represent the freezing during the first two and last two trials of the fear conditioning session. The trial blocks in the fear recall and extinction session and the extinction recall session represent freezing over blocks of two trials each (1 to 14).

## In-vivo electrophysiological recordings

All the animals were subjected to the recording of the local field potentials (LFPs) during tone habituation session, fear recall and extinction session and extinction recall session. Auditory-evoked potentials (AEPs) were recorded by connecting the microelectrodes to a unit gain buffer head stage (HS-36-Flex; Neuralynx, Bozeman, Montana, USA) and a data acquisition system Digilynx (Neuralynx, Bozeman, Montana, USA). Neural data were amplified (1000 times) and acquired at a sampling rate of 1 kHz followed by a band-pass filter (1–500 Hz) using Cheetah data acquisition software (Neuralynx, Bozeman, Montana, USA).

## Pharmacological infusion of muscimol into the basolateral amygdala

Intra-amygdalar infusion of muscimol was performed 30 min prior to fear recall and extinction session on Day 12. The infusion was performed using standard pressure injection methods. Infusion procedure was performed in the homecage. Injection cannulae with 1 mm projection (28 gauge, Plastic One, Roanoke, Virginia, USA) were inserted through the guide-cannulae. The injection cannula was connected to a Hamilton syringe (10 µl) using a polyethylene tubing (Plastic One, Roanoke, Virginia, USA), which was mounted on an infusion pump (Harvard Apparatus, Holliston, Massachusetts, USA). Rats were infused bilaterally with one hemisphere at a time. Muscimol (0.5 µl per side, 1 µg µl$^{-1}$ in saline; Tocris Biosciences, Bristol, UK) was infused at a rate of 0.1 µl min$^{-1}$. The injection cannula was taken out 5 min after the end of infusion, to allow the drug to diffuse into the tissue. After the completion of experiments, cannulae placement was confirmed using standard histological methods.

## Data analyses

### Auditory Evoked Potentials (AEPs)

AEPs were averaged over all the tone pips for the specified trial blocks. Averaged AEPs were quantified by measuring the amplitude (*Ghosh et al., 2013*). The amplitude was measured by the difference between the maxima (dot) after the onset of the response and the negative peak (arrow) (*Figure 2—figure supplement 1*). AEP amplitudes were calculated before conditioning (last two trials of tone habituation), fear recall (first two trials of fear recall and extinction) and extinction recall (first two trials of extinction recall). The AEP amplitudes for all the sessions were normalized as a percentage to the AEP amplitude before conditioning for each animal.

### Time frequency analyses

Event related changes in spectral power were evaluated by time-frequency analysis performed using continuous wavelet transformation (MATLAB) on the averaged AEPs. Complex Morlet wavelets were used to compute phase and the amplitude of evoked responses within a frequency range from 2 to 100 Hz in steps of 0.1 Hz (*Ghosh et al., 2013*). The bandwidth parameter and center frequency of the mother wavelet were 2 and 1 Hz respectively. Subsequently, the wavelet power of the time series was calculated and expressed in decibels. This was followed by z-score calculation for each frequency band across the time series. Baseline average power for the duration of 0 to −200 milliseconds was subtracted across all time points for each frequency bands. Tone evoked theta power was calculated over the duration of 0 to 250 milliseconds from tone onset for frequency from 2 to 12 Hz.

### Coherence and amplitude correlation

Coherence between the BLA and dmPFC was calculated using Welch's method (in built MATLAB function). The hamming window length used was 500 ms and 1,024 FFTs. Coherence was calculated over a wide frequency range. Theta coherence was quantified over 2–12 Hz frequency bandwidth.

For amplitude cross correlation (*Adhikari et al., 2010*), the signals were filtered between 2–12 Hz. Then instantaneous phase and amplitude of the signals were obtained using Hilbert transformation (in built MATLAB function). Next the mean amplitude of the signal was subtracted from the instantaneous amplitudes to remove the DC component. Then the cross-correlation between the amplitudes of the two signals was computed with over lags ranging from +0.1 to −0.1 s. The lead/lag at which the cross-correlation peaked was then determined. Then the dmPFC and BLA instantaneous amplitudes were randomly shifted by 2–5 s relative to each other 100 times. The shifted amplitudes were cross-correlated to find peaks expected by chance. The actual cross-correlation was considered significant if its peak value was greater than 95% of the peaks generated by

randomly shifted signal cross-correlations. Finally the distributions of these peaks were obtained and the time bin for the maxima of the distribution was quantified as the resultant lead/lag between the signals.

## Histology

After the experiment was concluded, rats were deeply anesthetized (ketamine/xylazine, 100/20 mg per kg). Electrolytic lesions (20 µA, 20 s) were made to mark the in vivo infusion sites. The animals were then perfused transcardially with ice-cold saline (0.9%) followed by 10% (vol/vol) formalin. The perfused brain was left in 10% (vol/vol) formalin overnight. Coronal sections (80 µm) were prepared using a vibratome (VT 1200S, Leica Microsystems, Wetzlar, Germany) and mounted on gelatin-coated glass slides. Sections were stained with 0.2% (wt/vol) cresyl violet solution and mounted with DPX (Sigma-Aldrich, St. Louis, Missouri, United States). The slides were imaged to identify and reconstruct infusion sites (*Figure 2a*).

## Statistical analyses

All values are expressed as mean ±SEM unless mentioned otherwise. Each data set was evaluated for outliers, which was defined as greater than twice the standard deviation away from the mean. Freezing within the tone habituation and conditioning session was evaluated using a one way repeated measures ANOVA followed by Tukey's post hoc test. Freezing in the fear recall and extinction as well as the extinction recall session was analyzed using two way repeated measures ANOVA followed by Holm-Sidak's post hoc test. For the AEP amplitudes and tone evoked theta powers, one way repeated measures ANOVA followed by Tukey's post hoc test was used to analyse the data for control and CIS groups separately. Next, to determine if the theta coherence and amplitude correlation lead/lags are significantly different than zero, unpaired Student's t test was used. For the behavior experiments with fear conditioning after CIS, two way repeated measures ANOVA was used to analyze the data for the tone habituation and conditioning session and the fear recall and extinction session. This was followed by Holm-Sidak's post hoc test. For comparing the freezing response during fear recall with/without BLA inactivation two way ANOVA was used followed by Holm-Sidak's post hoc test. All the data sets passed the normality test. Sample sizes were not determined prior to the experiments and were sufficient to get significant effects. All statistical tests were performed using GraphPad Prism (GraphPad software Inc., La Jolla, California, USA).

## Acknowledgements

We are grateful to Dr. Gregory Quirk (non-anonymous reviewer) and Dr. Joshua Gordon for helpful advice and discussions. This work was supported by funds from the Department of Atomic Energy and Department of Biotechnology, Government of India, and the Madan and Usha Sethi Fellowship.

## Additional information

### Funding

| Funder | Author |
| --- | --- |
| Department of Atomic Energy, Government of India | Mohammed Mostafizur Rahman Ashutosh Shukla Sumantra Chattarji |
| Department of Biotechnology, Ministry of Science and Technology | Sumantra Chattarji |
| Madan and Usha Sethi Fellowship | Sumantra Chattarji |

The funders had no role in study design, data collection and interpretation, or the decision to submit the work for publication.

## Author contributions
Mohammed Mostafizur Rahman, Conceptualization, Data curation, Formal analysis, Validation, Investigation, Visualization, Methodology, Writing—original draft, Writing—review and editing; Ashutosh Shukla, Formal analysis, Validation, Investigation; Sumantra Chattarji, Conceptualization, Resources, Data curation, Supervision, Funding acquisition, Methodology, Writing—original draft, Project administration, Writing—review and editing

## Author ORCIDs
Mohammed Mostafizur Rahman https://orcid.org/0000-0002-9355-4867
Sumantra Chattarji http://orcid.org/0000-0001-9962-3635

## Ethics
Animal experimentation: All animal care and experimentation procedures were approved by the Institutional Animal Ethics Committee, National Centre for Biological Sciences (Approval No: SC-5/2009) and Committee for the Purpose of Control and Supervision of Experiments on Animals, Government of India (Registration No: 109/CPCSEA).

## Decision letter and Author response
Decision letter https://doi.org/10.7554/eLife.35450.019
Author response https://doi.org/10.7554/eLife.35450.020

# Additional files

## Supplementary files
• Transparent reporting form
DOI: https://doi.org/10.7554/eLife.35450.017

## Data availability
Data generated or analysed during this study are included in the manuscript and supporting files. Source data files have been provided for all the Figures.

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
