## [Decision Letter]

Thank you for submitting your article "Extinction recall of fear memories formed before stress is not affected despite amygdalar hyperactivity" for consideration by *eLife*. Your article has been reviewed by three peer reviewers, including Jennifer L Raymond as the Reviewing Editor and Reviewer #1, and the evaluation has been overseen by Michael Frank as the Senior Editor. The following individual involved in review of your submission has agreed to reveal their identity: Gregory Quirk (Reviewer #2).

The reviewers have discussed the reviews with one another and the Reviewing Editor has drafted this decision to help you prepare a revised submission.

Summary:

This manuscript challenges the idea that stress impairs the extinction of fear memories. The authors replicate previous observations of impaired extinction when chronic stress occurs before fear conditioning, which had provided support for the idea that stress affects extinction. However, they report normal extinction and extinction recall in rats that underwent chronic restraint stress before extinction, but after fear conditioning. This dissociation of the effects of stress on acquisition vs. extinction of fear conditioning suggests that stress acts primarily on acquisition. Surprisingly, stress affected the CS (tone)-evoked field potential amplitude and theta power in the basolateral amygdala during extinction recall, although the behavior was normal. This raises the interesting question of what prevents the elevated BLA responses in the stressed animals from affecting behavior. Some clues are provided by recordings from the dorsal mPFC. In mPFC the theta power paralleled the behavior during fear recall and extinction recall in the stressed and unstressed animals, although the evoked potentials during fear recall were altered by stress. In addition, stress blocked the enhanced theta mPFC and the BLA that is observed in the unstressed rates during fear recall and extinction. The paper is very accessible and the figures are clear and convincing. Analysis of the effects of chronic stress on fear extinction, independent of stress's effects on fear conditioning, is long overdue, given the developing story that chronic stress supposedly impairs extinction. Impact on the field should be significant.

Essential revisions:

1) Substantial revision of the text is required to more carefully interpret the results, and more fully consider how previous work could affect the interpretation.

a) The behavioral hypothesis stated in the Introduction and Discussion is that earlier studies induced stress prior to conditioning, which caused rats to start extinction at higher levels of freezing, which would then take longer to extinguish and appear as an extinction deficit. However, this is not exactly what the authors show in their replication of the pre-conditioning stress experiment (Figure 1—figure supplement 1). The stressed rats seem to start extinction only slightly higher than controls, but then show absolutely no within-session extinction, and impaired recall the following day. This suggests that the effects of stress on BLA excitability and theta that they observed causes conditioning to occur in a way that resists subsequent extinction (even though expression of fear is not elevated that much). The authors may wish to speculate on this, which may be more interesting than simply starting extinction from a higher freezing level.

The following statement would especially benefit from more careful wording: "stressed animals had to extinguish fear memories that were invariably stronger than unstressed animals" The metric for "invariably stronger" is not clear, since such as Noble et al. (2017) show that this statement is not true if one's metric of "stronger" is% freezing during fear recall. If, on the other hand, one's metric of "stronger" is resistance to extinction, the authors may be correct, but probably don't have enough data yet to claim "inevitably."

b) Discuss the potential reasons for the different outcomes of pre- and post-conditioning stress on extinction. One possibility is that the encoding of fear is different in stressed animals so that extinction operates on this in a different manner. Another possibility is that the circuits supporting fear recall are different at different times after acquisition, and therefore undergo extinction differently – this could come into play in the current study, since the time between acquisition and extinction was different for the pre- and post-conditioning stress conditions (see point #2 below).

c) Previous studies showing that chronic post-conditioning stress doesn't have any impact on the recall of fear memories acquired prior to stress should be discussed; see Meyer et al. (2014).

d) Results, second paragraph. It is not clear why the authors are equating high levels of pip-evoked theta power with "stress-induced hyperactivity". High levels of pip-evoked theta do not necessarily reflect hyperactivity or hyperexcitability in neurons. In this regard, the title, which refers to hyperactivity rather than theta, is misleading.

e) The deficit in fear extinction in previously stressed rats in Zhang and Rosenkranz was only observed in adolescent rats. No effect of stress on extinction was observed in adult rats. It is this latter finding that is most relevant to the findings in the manuscript. This point should be highlighted in the current manuscript.

2) It is convenient that post-conditioning stress did not increase the expression of freezing, so that extinction could be objectively and cleanly assessed. However, this raises a question: if BLA is so important for expression of conditioned fear, why was expression of freezing not increased, especially with the observed hyperexcitability in BLA? One (unlikely?) possibility is that the circuits within BLA affected by stress were separate from the circuits that were previously conditioned. Another possibility is that BLA is not needed for expression of conditioned fear at that timepoint. This (somewhat heretical) idea comes from a 2015 study showing that optogenetic silencing of BLA reduced freezing when it was performed 6 hrs, but not 7 days, after auditory fear conditioning (Do-Monte et al., 2015, Extended Data Figure 8). Because post-conditioning stress was carried out for 10 days, this may have outlasted BLA involvement in expression of freezing, and could explain why excitability changes were not reflected in freezing at that timepoint. To test this hypothesis, the authors could inactivate BLA at that timepoint to determine the extent to which BLA is (or is not) necessary for expression of older fear memories. Alternatively, they could suggest this possibility in the Discussion.

3) Figure 1 shows a slower the time course of extinction in the stressed rats. Are there parallels in the physiology? More generally, it would be helpful to see a time course in Figures 2 and 3.

4) In the Figure 2 legends for panels E and F, and in Figure 3, it is not clear how the degrees of freedom were calculated. These should be integer values, not values with decimal places. Please clarify this.

[Editors' note: further revisions were requested prior to acceptance, as described below.]

Thank you for submitting your article "Extinction recall of fear memories formed before stress is not affected despite higher theta activity in the amygdala" for consideration by *eLife*. Your article has been reviewed by two peer reviewers, and the evaluation has been overseen by Jennifer Raymond as the Reviewing Editor and Michael Frank as the Senior Editor. The following individual involved in review of your submission has agreed to reveal their identity: Gregory Quirk (Reviewer #2).

The reviewers have discussed the reviews with one another and the Reviewing Editor has drafted this decision to help you prepare a revised submission.

Summary:

This manuscript provides an analysis of the effects of chronic stress on fear extinction, independent of the effects of stress on fear conditioning. Whereas the prevailing view in the field has been that stress impairs the extinction of fear memories, the authors report normal extinction and extinction recall in rats that underwent chronic restraint stress before extinction, but after fear conditioning. This finding should have significant scientific impact and may also have substantial clinical impact. The reviewers are enthusiastic about the manuscript, and agreed that the new experiments included in the revised submission have the potential to add considerably to the story. However, the reviewers also felt those experiments were difficult to interpret without a positive control for the effective pharmacological inhibition of BLA.

Essential revision:

The reviewers agree that the manuscript is enhanced by the new experiments showing that pharmacological inactivation of the BLA 12 days after conditioning had no effect on fear expression or extinction. This new finding explains a conundrum from the previous version, namely, why was fear expression normal in recently stressed rats even though BLA excitability was increased? The answer seems to be that BLA is not involved in fear expression at that timepoint. This increases our understanding and interpretation of chronic stress experiments: not only is the timing relative to conditioning important, but so is the duration of the stress period, as BLA's involvement is time-limited.

The reviewers were impressed that the authors provided new experiments in response to the request in the initial review to simply "Discuss the potential reasons for the different outcomes of pre- and post-conditioning stress on extinction." However, given the negative result (no effect of BLA inactivation on fear expression at the late time point), a positive control is needed to show that the pharmacological inactivation of BLA was successful, by showing that BLA inactivation in the same place/dose but at an earlier time point impairs memory expression whereas at a later time it does not (with histological placements of cannula tips shown to confirm location in BLA and the volume and concentration of muscimol provided). The reviewers did not feel comfortable with the inclusion of the new experiments in the manuscript without this essential control, and hope that the authors will choose to do these additional experiments.

Alternatively, since the authors had gone above-and-beyond the reviewers' request to simply "discuss" potential reasons, an acceptable alternative would be to remove the new experiments, and simply discuss potential reasons based on the published results in the literature.

---

## [Author Response]

Essential revisions:1) Substantial revision of the text is required to more carefully interpret the results, and more fully consider how previous work could affect the interpretation.a) The behavioral hypothesis stated in the Introduction and Discussion is that earlier studies induced stress prior to conditioning, which caused rats to start extinction at higher levels of freezing, which would then take longer to extinguish and appear as an extinction deficit. However, this is not exactly what the authors show in their replication of the pre-conditioning stress experiment (Figure 1—figure supplement 1). The stressed rats seem to start extinction only slightly higher than controls, but then show absolutely no within-session extinction, and impaired recall the following day. This suggests that the effects of stress on BLA excitability and theta that they observed causes conditioning to occur in a way that resists subsequent extinction (even though expression of fear is not elevated that much). The authors may wish to speculate on this, which may be more interesting than simply starting extinction from a higher freezing level.The following statement would especially benefit from more careful wording: "stressed animals had to extinguish fear memories that were invariably stronger than unstressed animals" The metric for "invariably stronger" is not clear, since such as Noble et al. (2017) show that this statement is not true if one's metric of "stronger" is% freezing during fear recall. If, on the other hand, one's metric of "stronger" is resistance to extinction, the authors may be correct, but probably don't have enough data yet to claim "inevitably."

We thank the reviewer for raising this important point. The reviewer rightly points out that our original text implied that the metric for “stronger” fear memory is higher % freezing. However, in our pre-conditioning stress experiments, exposure to chronic stress caused conditioning to occur in a way that resists subsequent extinction even though expression of fear is not significantly higher to start with(data originally shown in Figure 1—figure supplement 1, now moved to main Figure 1C-F). Hence, we agree that resistance to extinction offers a metric that better reflects our data. Accordingly, we have revised the Introduction to incorporate this suggestion by the reviewer (last paragraph). We have also removed the term “invariably” from the Introduction. Both the Introduction (last paragraph) and Results and Discussion (fifth paragraph) have been modified to elaborate on this point. These sections have also been revised to mention the references to earlier studies relevant to these points, as advised by the reviewer.

b) Discuss the potential reasons for the different outcomes of pre- and post-conditioning stress on extinction. One possibility is that the encoding of fear is different in stressed animals so that extinction operates on this in a different manner. Another possibility is that the circuits supporting fear recall are different at different times after acquisition, and therefore undergo extinction differently – this could come into play in the current study, since the time between acquisition and extinction was different for the pre- and post-conditioning stress conditions (see point #2 below).

We thank the reviewer for this suggestion. We have now revised the Results and Discussion to elaborate on the potential reasons underlying the different outcomes of pre- versus post-conditioning stress on fear extinction (Results and Discussion, seventh paragraph). Specifically, the new data we have added from our analysis of the effects of muscimol inactivation of the BLA (new Figure 4) shed light on the different possibilities suggested by the reviewer (see response to point #2 below). In addition to these new results, we have also added a new figure (Figure 5) to summarize all of these points. And we have devoted a new paragraph in the Results and Discussion section to elaborate on the key points depicted in Figure 5.

c) Previous studies showing that chronic post-conditioning stress doesn't have any impact on the recall of fear memories acquired prior to stress should be discussed; see Meyer et al. (2014).

We thank the reviewer for drawing our attention to the study by Meyer et al. (2014). We apologize for this omission and have now modified the Results and Discussion section to include the relevant findings reported in the earlier study (Results and Discussion, fifth paragraph).

d) Results, second paragraph. It is not clear why the authors are equating high levels of pip-evoked theta power with "stress-induced hyperactivity". High levels of pip-evoked theta do not necessarily reflect hyperactivity or hyperexcitability in neurons. In this regard, the title, which refers to hyperactivity rather than theta, is misleading.

We have modified the title to in response to the concern expressed by the reviewer:

“Extinction recall of fear memories formed before stress is not affected despite amygdalar theta hyperactivity”.

We have revised the Abstract to address this concern.

Finally, we have modified the Results and Discussion section, wherever relevant, to clarify this point.

e) The deficit in fear extinction in previously stressed rats in Zhang and Rosenkranz was only observed in adolescent rats. No effect of stress on extinction was observed in adult rats. It is this latter finding that is most relevant to the findings in the manuscript. This point should be highlighted in the current manuscript.

We thank the reviewer for pointing this out. We have now mentioned this finding in the revised Results and Discussion section (fifth paragraph).

2) It is convenient that post-conditioning stress did not increase the expression of freezing, so that extinction could be objectively and cleanly assessed. However, this raises a question: if BLA is so important for expression of conditioned fear, why was expression of freezing not increased, especially with the observed hyperexcitability in BLA? One (unlikely?) possibility is that the circuits within BLA affected by stress were separate from the circuits that were previously conditioned. Another possibility is that BLA is not needed for expression of conditioned fear at that timepoint. This (somewhat heretical) idea comes from a 2015 study showing that optogenetic silencing of BLA reduced freezing when it was performed 6 hrs, but not 7 days, after auditory fear conditioning (Do-Monte et al., 2015, Extended Data Figure 8). Because post-conditioning stress was carried out for 10 days, this may have outlasted BLA involvement in expression of freezing, and could explain why excitability changes were not reflected in freezing at that timepoint. To test this hypothesis, the authors could inactivate BLA at that timepoint to determine the extent to which BLA is (or is not) necessary for expression of older fear memories. Alternatively, they could suggest this possibility in the Discussion.

This point raised by the reviewer is a very interesting one indeed. Some of the issues mentioned in point #1 also center on this question of why theta hyperactivity in the BLA, despite its pivotal role in the expression of conditioned fear, is not manifested as enhanced freezing in the stressed animals. The reviewer suggests two possible explanations for our findings. The more likely, and particularly intriguing (and “somewhat heretical”, according to the reviewer), possibility is based on a study by Do-Monte et al. (2015) that showed that the BLA is not needed for expression of conditioned fear 7 days after auditory fear conditioning. Since our chronic stress protocol lasts for 10 days, it is possible that BLA hyperactivity no longer affects the expression of the older fear memories. Hence, as suggested by the reviewer, we carried out new experiments to test this possibility (instead of just elaborating on this is in the Discussion) using in vivo infusion of muscimol directly into the BLA just before fear recall and extinction on Day 12 (Figure 4). Strikingly, these new experiments confirm that fear expression at this time point no longer depends on the BLA (in both stressed and control rats). These results are now described in a new figure (Figure 4). We added a new Figure 5 to discuss the implications of these new findings.

3) Figure 1 shows a slower the time course of extinction in the stressed rats. Are there parallels in the physiology? More generally, it would be helpful to see a time course in Figures 2 and 3.

As advised by the reviewer, we have carried out analyses of our electrophysiological recordings to examine if these in vivo measurements parallel the time course of gradual changes in freezing behavior. These additional analyses revealed that the gradual decrease in dmPFC theta power was a reliable electrophysiological correlate of the significant within-session reduction in freezing during the acquisition of extinction (Day 12, Figure 1B) in both control and stressed animals (new Figure 2—figure supplement 2). However, the slower time course of extinction in the stressed rats was not reflected as a difference in the time course of reduction in dmPFC theta power between stressed and control animals. Similar analysis of BLA theta power during extinction learning (Day 12, Figure 1B) did not reveal any significant differences that paralleled the gradual within-session decrease in freezing levels (new Figure 2—figure supplement 2). Though we detected a decreasing trend in BLA CS-evoked theta power by trial block 4 during extinction learning (Day 12), this was not statistically significant. In contrast, no such change was evident in stressed animals. Finally, the stress-induced disruption in both the dmPFC-BLA theta synchrony and dmPFC-to-BLA directional influence is evident across all trial blocks during the acquisition of extinction the previous day (Figure 3—figure supplement 1). Hence, the absence of any changes in these measures across trial blocks meant that they were not good indicators of the gradual and significant reduction in within-session freezing during extinction learning.

These results are now reported in Figure 2—figure supplement 2 and Figure 3—figure supplement 1, as well as in the main text (Results and Discussion, second and third paragraphs).

4) In the Figure 2 legends for panels E and F, and in Figure 3, it is not clear how the degrees of freedom were calculated. These should be integer values, not values with decimal places. Please clarify this.

We used repeated measures ANOVA to analyze the results. While this design reduces error in variance due to individuals between groups, it also introduces a correlation of errors in individuals between groups. This happens because the same individual is tested across groups. At times this causes a violation of assumptions of sphericity of data. Since sphericity of data is one of the assumptions for ANOVA, its violation needs to be corrected using Greenhouse-Geisser correction (Abdi, Encyclopedia of research design 1, 2010). Assuming the data set might violate the assumptions of sphericity, we applied the Greenhouse-Geisser correction. However, we agree that with the sample size we have, it is very difficult to conclude if the assumptions of sphericity are violated. Our earlier analysis adopted a cautious approach. Hence, we have now modified our analysis. Hence, we no longer assume violation of assumptions of sphericity and not use the Greenhouse-Geisser correction. Therefore, the degrees of freedom are integers and not values with decimal places now. This is consistent with the suggestion made by the reviewer. We have rectified this in Figure 3 and the corresponding legends. Importantly, this change in analysis does not affect the interpretation of any of our results.

[Editors' note: further revisions were requested prior to acceptance, as described below.]

Essential revision:The reviewers agree that the manuscript is enhanced by the new experiments showing that pharmacological inactivation of the BLA 12 days after conditioning had no effect on fear expression or extinction. This new finding explains a conundrum from the previous version, namely, why was fear expression normal in recently stressed rats even though BLA excitability was increased? The answer seems to be that BLA is not involved in fear expression at that timepoint. This increases our understanding and interpretation of chronic stress experiments: not only is the timing relative to conditioning important, but so is the duration of the stress period, as BLA's involvement is time-limited.The reviewers were impressed that the authors provided new experiments in response to the request in the initial review to simply "Discuss the potential reasons for the different outcomes of pre- and post-conditioning stress on extinction." However, given the negative result (no effect of BLA inactivation on fear expression at the late time point), a positive control is needed to show that the pharmacological inactivation of BLA was successful, by showing that BLA inactivation in the same place/dose but at an earlier time point impairs memory expression whereas at a later time it does not (with histological placements of cannula tips shown to confirm location in BLA and the volume and concentration of muscimol provided). The reviewers did not feel comfortable with the inclusion of the new experiments in the manuscript without this essential control, and hope that the authors will choose to do these additional experiments.Alternatively, since the authors had gone above-and-beyond the reviewers' request to simply "discuss" potential reasons, an acceptable alternative would be to remove the new experiments, and simply discuss potential reasons based on the published results in the literature.

We are grateful to the reviewers for their flexibility in how this issue can be addressed in our revised manuscript. We agree with the reviewers and appreciate the value of having a positive control confirming that pharmacological inactivation of BLA at an earlier time point is successful in impairing fear memory expression. Indeed, it is the importance of directly testing why BLA hyperactivity in stressed animals did not affect fear expression that prompted us to do the new inactivation experiments. This, in turn, led to the new question about a positive control raised by the reviewers.

Unfortunately, Dr. Rahman, the first author who carried out all of the in vivo experiments for this study, has already left the lab to start his post-doctoral work abroad. As a result, we do not have anybody else in the lab at this time who can complete these additional control experiments in a timely fashion. Further, the elegant study by Do-Monte et al. (Nature, 2015), showing optogenetic silencing of BLA reduced freezing when it was performed 6 hours, but not 7 days, after auditory fear conditioning, addresses the same issue. So, we hope this will partly make up for the absence of a positive control confirming the same finding using a pharmacological inactivation in the present study. Thus, as much as we would like to do these control experiments, we are unable to do so due to constraints that are beyond our control. Please accept our sincere apologies for this.

Hence, we have removed the findings on the BLA inactivation experiments (Figure 4 in the earlier version) from the Results and Discussion section. Instead, we have opted for the “acceptable alternative” by addressing these points in the revised Results and Discussion section. Further, since the reviewers have expressed positive views on the overall utility of the final summary figure (new Figure 4, which was Figure 5 in the earlier version), we have covered these points in the form of preliminary results in Figure 4—figure supplement 1.